# Epitomic Variational Autoencoder

**Serena Yeung** *
Stanford University
{serena}@cs.stanford.edu

**Anitha Kannan & Yann Dauphin**
Facebook AI Research
{akannan,ynd}@fb.com

**Li Fei-Fei**
Stanford University
{feifeili}@cs.stanford.edu

## ABSTRACT

In this paper, we propose *epitomic variational autoencoder* (eVAE), a probabilistic generative model of high dimensional data. eVAE is composed of a number of sparse variational autoencoders called 'epitome' such that each epitome partially shares its encoder-decoder architecture with other epitomes in the composition. We show that the proposed model greatly overcomes the common problem in variational autoencoders (VAE) of model over-pruning. We substantiate that eVAE is efficient in using its model capacity and generalizes better than VAE, by presenting qualitative and quantitative results on MNIST and TFD datasets.

## 1 INTRODUCTION

Unsupervised learning holds the promise of learning the inherent structure in data so as to enable many future tasks including generation, prediction and visualization. Generative modeling is an approach to unsupervised learning wherein an explicit stochastic generative model of data is defined, such that independent draws from this model are likely to produce the original data distribution, while the learned latent structure itself is useful in prediction, classification and visualization tasks.

The recently proposed variational autoencoder (VAE) (Kingma & Welling, 2014) is an example of one such generative model. VAE pairs a top down generative model with a bottom up recognition network for amortized probabilistic inference. Both networks are jointly trained to maximize a variational lower bound on the data likelihood. A number of recent works use VAE as a modeling framework, including iterative conditional generation of images (Gregor et al., 2015) and conditional future frame prediction (Xue et al., 2016).

A commonly known problem with the VAE lower bound is that it is known to self-prune or under utilize the model's capacity (Mackay, 2001). This can lead to poor generalization. A common approach to alleviate this problem is to resort to optimization schedules and regularization techniques (Bowman et al., 2015; Kaae Sonderby et al., 2016) that trade-off two competing terms, latent cost and data reconstruction, in the bound. Fig. 1 provides a quick insight into this problem of over-pruning and how commonly used regularization techniques may not be sufficient. Detailed discussion is provided in § 2.1.

In this paper, we take a model-based approach to directly address this problem. We present an extension of variational autoencoders called epitomic variational autoencoder (Epitomic VAE, or eVAE, for short) that automatically learns to utilize its model capacity more effectively, leading to better generalization. Consider the task of learning a $D$-dimensional representation for the examples in a given dataset. The motivation for our model stems from the hypothesis that a single example in the dataset can be sufficiently embedded in a smaller $K$-dimensional ($K \ll D$) subspace of $D$. However, different data points may need different subspaces, hence the need for $D$. Sparse coding methods also exploit a similar hypothesis. Epitomic VAE exploits sparsity using an additional categorical latent variable in the encoder-decoder architecture of the VAE. Each value of the variable activates only a contiguous subset of latent stochastic variables to generate an observation. This

---

*Work done during an internship at Facebook AI Research.

enables learning multiple shared subspaces such that each subspace specializes, and also increases the use of model capacity (Fig. 4), enabling better representation. The choice of the name *Epitomic VAE* comes from the fact that multiple miniature models with shared parameters are trained simultaneously.

The rest of the paper is organized as follows. We first describe variational autoencoders and mathematically show the model pruning effect in § 2. We then present our epitomic VAE model in § 3 that overcomes these shortcomings. Experiments showing qualitative and quantitative results are presented in § 4. We finally provide more general context of our work in the related work in § 5, and conclude with discussions.

## 2 VARIATIONAL AUTOENCODERS

The generative model (decoder) of a VAE consists of first generating a D-dimensional stochastic variable $\mathbf{z}$ drawn from a standard multivariate Gaussian

$$p(\mathbf{z}) = \mathcal{N}(\mathbf{z}; 0; I) \tag{1}$$

and then generating the N-dimensional observation $\mathbf{x}$ from a parametric family of distributions such as a Gaussian

$$p_\theta(\mathbf{x}|\mathbf{z}) = \mathcal{N}(\mathbf{x}; f_1(\mathbf{z}); \exp(f_2(\mathbf{z}))) \tag{2}$$

where $f_1$ and $f_2$ define non-linear deterministic transformations of $\mathbf{z}$ modeled using a neural network. The parameters $\theta$ of the model are the weights and biases of the neural network that encodes the functions $f_1$ and $f_2$.

Given a dataset $X$ of $T$ *i.i.d* samples, the model is learned such that it maximizes the likelihood of the parameters to have generated the data, $p(X|\theta)$. This maximization requires marginalizing the unobserved $\mathbf{z}$. However, computing $p(\mathbf{z}|\mathbf{x})$ is intractable due to dependencies induced between the $z_i$ when conditioned on $\mathbf{x}$.

Variational autoencoders, as the name suggests, use variational inference to approximate the exact posterior with a surrogate parameterized distribution. However, instead of having separate parameters for the posterior distribution of each observation, VAE amortizes the cost by learning a neural network with parameters $\phi$ that outputs the posterior distribution of the form $q_\phi(\mathbf{z}|\mathbf{x}) = \prod_d q(z_i|\mathbf{x})$. This results in the lower bound given by

$$\log p_\theta(X) = \sum_{t=1}^{T} \log \int_{\mathbf{z}} p_\theta(\mathbf{x}^{(t)}, \mathbf{z}) \tag{3}$$

$$\geq \sum_{t=1}^{T} E_{q_\phi(\mathbf{z}|\mathbf{x}^{(t)})}[\log p(\mathbf{x}^{(t)}|\mathbf{z})] - KL\Big(q_\phi(\mathbf{z}|\mathbf{x}^{(t)}) \parallel p(\mathbf{z})\Big) \tag{4}$$

VAE is trained with standard backpropagation using minibatch gradient descent to minimize the negative of the lowerbound

$$\mathcal{C}_{vae} = -\sum_{t=1}^{T} E_{q_\phi(\mathbf{z}|\mathbf{x}^{(t)})}[\log p(\mathbf{x}^{(t)}|\mathbf{z})] + \sum_{t=1}^{T} \sum_{i=1}^{D} KL\Big(q_\phi(z_i|\mathbf{x}^{(t)}) \parallel p(z_i)\Big) \tag{5}$$

### 2.1 AUTOMATIC MODEL OVER-PRUNING IN VAE

$\mathcal{C}_{vae}$ introduces a trade-off between data reconstruction (first term) and satisfying the independence assumption of $p(\mathbf{z})$ (second term, KL).

Of particular interest is the KL term. Since the KL term is the sum of independent contributions from each dimension $d$ of $D$, it provides unduly freedom for the model in how it minimizes this term. In particular, the model needs to only ensure that the overall KL term is minimized, on average, and not per component wise. The easiest way for the model to do this is to have a large number of components that satisfies the KL term effectively, by turning off the units so that the posterior for those units becomes the same as the prior[1]. This effect is quite pronounced in the early iterations of

---

[1]Since log variance is modeled using the neural network, turning it off will lead to a variance of 1.

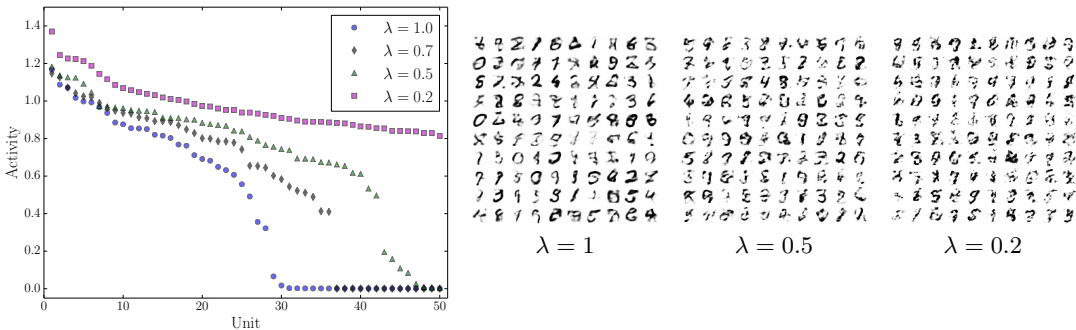

Figure 1: Sorted activity level of latent units and corresponding generations on MNIST, for a 50-d VAE with a hidden layer of 500 units. Shown for varying values of the KL weight $\lambda$. When $\lambda = 1$, only 30 units are active. As $\lambda$ is decreased, more units are active; however generation does not improve since the model uses the capacity to model increasingly well only regions of the posterior manifold near training samples (see reconstructions in Fig. 8).

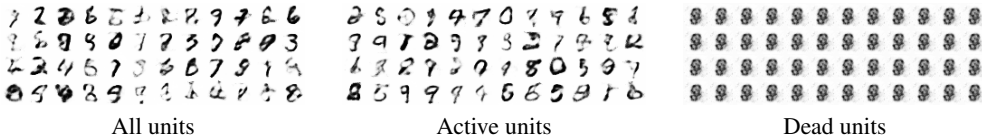

Figure 2: Only active units contribute to generation, whereas units that have "died" have no effect. Shown for a 50-d VAE with $\lambda = 1$.

training: the model for $\log p(x|z)$ is quite impoverished and hence the easiest way to improve the bound is by turning off the KL terms. However, once the units have become inactive, it is almost impossible for them to resurrect, and hence the full capacity of the model is not utilized.

A quantity that is useful in understanding this effect, is the activity level of a unit. Following Burda et al. (2015), we define a unit to be used, or "active", if $A_u = \text{Cov}_x(\mathbb{E}_{u \sim q(u|\mathbf{x})}[u]) > 0.02$.

A commonly used approach to overcome this problem is to use a trade-off between the two terms using parameter $\lambda$ so that the cost is

$$\mathcal{C} = -E_{q_\phi(\mathbf{z}|\mathbf{x})}[\log p(\mathbf{x}|\mathbf{z})] + \lambda \sum_{i=1}^{D} KL\Big(q_\phi(z_i|\mathbf{x}) \parallel p(z_i)\Big) \tag{6}$$

Fig. 1 shows the effect of $\lambda$ on unit activity and generation, with $\lambda = 1$ being the correct objective to optimize. While tuning down $\lambda$ increases the number of active units, samples generated from the model are still poor. Fig. 2 shows generation using all units, active units only, and dead units only, for $\lambda = 1$. The model spends its capacity in ensuring that reconstruction of the training set is optimized (reconstruction visualizations are shown in § 8.1), at the cost of generalization. This has led to more sophisticated schemes such as using an annealed optimization schedule for $\lambda$ (Bowman et al., 2015; Kaae Sonderby et al., 2016) or enforcing minimum KL contribution from subsets of the latent units (Kingma et al., 2016).

In this paper, we present a model based approach called "epitomic variational autoencoder" to address the problem of over pruning.

## 3 MODEL

We propose epitomic variational autoencoder (eVAE) to overcome the shortcomings of VAE by enabling more efficient use of model capacity to gain better generalization. We base this on the observation that while we may need a $D$-dimensional representation to accurately represent every example in a dataset, each individual example can be represented with a smaller $K$-dimensional subspace. As an example, consider MNIST with its variability in terms of digits, strokes and thick-

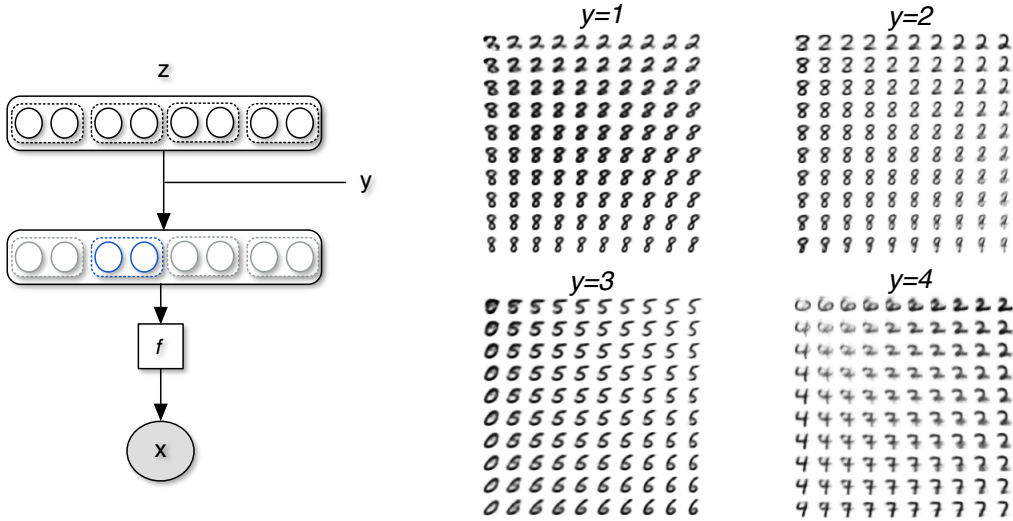

Figure 3: Left: Illustration of an epitomic VAE with dimension D=8, epitome size K=2 and stride S=2. In this depiction, the second epitome is active. Right: Learned manifolds on MNIST for 4 different epitomes in a 20-d eVAE with size $K = 2$ and stride $s = 1$. We observe that each epitome specializes on a coherent subset of examples.

ness of ink, to name a few. While the overall $D$ is large, it is likely that only a few $K$ dimensions of $D$ are needed to capture the variability in strokes of some digits (see Fig. 3).

Epitomic VAE can be viewed as a variational autoencoder with latent stochastic dimension $D$ that is composed of a number of sparse variational autoencoders called *epitomes*, such that each epitome partially shares its encoder-decoder architecture with other epitomes in the composition. In this paper, we assume simple structured sparsity for each epitome: in particular, only $K$ *contiguous* dimensions of $D$ are active[2].

The generative process can be described as follows: A D-dimensional stochastic variable $\mathbf{z}$ is drawn from a standard multivariate Gaussian $p(\mathbf{z}) = \mathcal{N}(\mathbf{z}; 0; I)$. In tandem, an epitome is implicitly chosen through an epitome selector variable $y$, which has a uniform prior over possible epitomes. The $N$-dimensional observation $\mathbf{x}$ is then drawn from a Gaussian distribution:

$$p_\theta(\mathbf{x}|y, \mathbf{z}) = \mathcal{N}(\mathbf{x}; f_1(\mathbf{m}_y \odot \mathbf{z}), \exp(f_2(\mathbf{m}_y \odot \mathbf{z}))) \qquad (7)$$

$\mathbf{m}_y$ enforces the epitome constraint: it is also a $D$-dimensional vector that is zero everywhere except in the active dimensions of the epitome. $\odot$ is element-wise multiplication between the two operands. Thus, $\mathbf{m}_y$ masks the dimensions of $\mathbf{z}$ other than those dictated by the choice of $y$. Fig. 3 illustrates this for an 8-d $\mathbf{z}$ with epitome size $K = 2$, so that there are four possible epitomes (the model also allows for overlapping epitomes, but this is not shown for illustration purposes). Epitome structure is defined using size $K$ and stride $s$, where $s = 1$ corresponds to full overlap in $D$ dimensions[3]. Our model generalizes the VAE and collapses to a VAE when $D = K = s$.

$f_1(\diamond)$ and $f_2(\diamond)$ define non-linear deterministic transformations of $\diamond$ modeled using neural networks. Note that the model does not snip off the $K$ dimensions corresponding to an epitome, but instead deactivates the D-K dimensions that are not part of the chosen epitome. While the same deterministic functions $f_1$ and $f_2$ are used for any choice of epitome, the functions can still specialize due to the

---

[2]The model also allows for incorporating other forms of structured sparsity.

[3]The strided epitome structure allows for learning $O(D)$ specialized subspaces, that when sampled during generation can each produce good samples. In contrast, if only a simple sparsity prior is introduced over arbitrary subsets (e.g. with Bernoulli latent units to specify if a unit is active for a particular example), it can lead to poor generation results, which we confirmed empirically but do not report. The reason for this is as follows: due to an exponential number of potential combinations of latent units, sampling a subset from the prior during generation cannot be straightforwardly guaranteed to be a good configuration for a subconcept in the data, and often leads to uninterpretable samples.

sparsity of their inputs. Neighboring epitomes will have more overlap than non-overlapping ones, which manifests itself in the representation space; an intrinsic ordering in the variability is learned.

## 3.1 OVERCOMING OVER-PRUNING

Following Kingma & Welling (2014), we use a recognition network $q(\mathbf{z}, y|\mathbf{x})$ for approximate posterior inference, with the functional form

$$
\begin{align}
q(\mathbf{z}, y|\mathbf{x}) &= q(y|\mathbf{x})q(\mathbf{z}|y, \mathbf{x}) \tag{8} \\
&= q(y|\mathbf{x})\mathcal{N}(\mathbf{z}; \mathbf{m}_y \odot \mu, \exp(\mathbf{m_y} \odot \phi)) \tag{9}
\end{align}
$$

where $\mu = \mathbf{h_1}(\mathbf{x})$ and $\phi = \mathbf{h_2}(\mathbf{x})$ are neural networks that map $\mathbf{x}$ to $D$ dimensional space.

We use a similar masking operation to deactivate units, as decided by the epitome $y$. Unlike the generative model (eq. 7), the masking operation defined by $y$ operates directly on outputs of the recognition network that characterizes the parameters of $q(\mathbf{z}|y, \mathbf{x})$.

As in VAE, we can derive the lower bound on the log probability of a dataset, and hence the cost function (negative bound) is

$$
\begin{align}
\mathcal{C}_{evae} &= -\sum_{t=1}^{T} E_{q(\mathbf{z}, y|\mathbf{x}^{(t)})}[\log p(\mathbf{x}^{(t)}|y, \mathbf{z})] \\
&\quad -\sum_{t=1}^{T} KL\left[q_\phi(y|\mathbf{x}^{(t)}) \parallel p_\theta(y)\right] - \sum_{t=1}^{T}\sum_{y} q_\phi(y|\mathbf{x}^{(t)})KL\left[q_\phi(\mathbf{z}|y, \mathbf{x}^{(t)}) \parallel p_\theta(\mathbf{z})\right] \tag{10}
\end{align}
$$

The epitomic VAE departs from the VAE in how the contribution from the KL term is constrained. Let us consider the third term in eq. 10, and substituting in eq. 9:

$$
\sum_{t=1}^{T}\sum_{y} q_\phi(y|\mathbf{x}^{(t)})KL\left[q_\phi(\mathbf{z}|y, \mathbf{x}^{(t)}) \parallel p_\theta(\mathbf{z})\right] \tag{11}
$$

$$
= \sum_{t=1}^{T}\sum_{y} q_\phi(y|\mathbf{x}^{(t)})KL\left[\mathcal{N}(\mathbf{z}; \mathbf{m}_y \odot \mu^{(\mathbf{t})}, \exp(\mathbf{m_y} \odot \phi^{(\mathbf{t})})) \parallel \mathcal{N}(\mathbf{z}; \mathbf{0}, \mathbf{I})\right] \tag{12}
$$

$$
= \sum_{t=1}^{T}\sum_{y} q_\phi(y|\mathbf{x}^{(t)}) \sum_{d=1}^{D} \mathbf{1}[m_{d,y} = 1]KL\left[\mathcal{N}(z_d; \mu_d^{(t)}, \exp(\phi_d^{(t)})) \parallel \mathcal{N}(0, 1)\right] \tag{13}
$$

where $\mathbf{1}[\star]$ is an indicator variable that evaluates to 1 if only if its operand $\star$ is true.

For a training example $\mathbf{x}^{(t)}$ and for a fixed $y$ (and hence the corresponding epitome), the number of KL terms that will contribute to the bound is exactly $K$. The dimensions of $z$ that are not part of the corresponding epitome will have zero KL because their posterior parameters are masked to have unit Gaussian, the same as the prior. By design, this ensures that only the $K$ dimensions that explain $\mathbf{x}^{(t)}$ contribute to $\mathcal{C}_{evae}$.

This is quite in contrast to how VAE optimizes $\mathcal{C}_{vae}$ (§. 2.1). For $\mathcal{C}_{vae}$ to have a small contribution from the KL term of a particular $z_d$, it has to infer that unit to have zero mean and unit variance for many examples in the training set. In practice, this results in VAE completely deactivating units, and leading to many dead units. EpitomicVAE chooses the epitome based on $\mathbf{x}^{(t)}$ and ensures that the dimensions that are not useful in explaining $\mathbf{x}^{(t)}$ are ignored in $\mathcal{C}_{evae}$. This means that the unit is still active, but by design, only a fraction of examples in the training set contributes a possible non-zero value to $z_d$'s KL term in $\mathcal{C}_{evae}$. This added flexibility gives the model the freedom to use more total units without deactivating them, while optimizing the bound. With these characteristics, during training, the data points will naturally group themselves to different epitomes, leading to a more balanced use of $\mathbf{z}$.

In Fig. 4 we compare the activity levels of VAE, dropout VAE and our model. We see that compared with VAE, our model is able to better use the model capacity. In the same figure, we also compare with adding dropout to the latent variable $\mathbf{z}$ of the VAE (Dropout VAE). While this increases the number of active units, it generalizes poorly as it uses the dropout layers to merely replicate representation, in contrast to eVAE. See Fig. 5 along with the explanation in § 4.1 where we compare generation results for all three models.

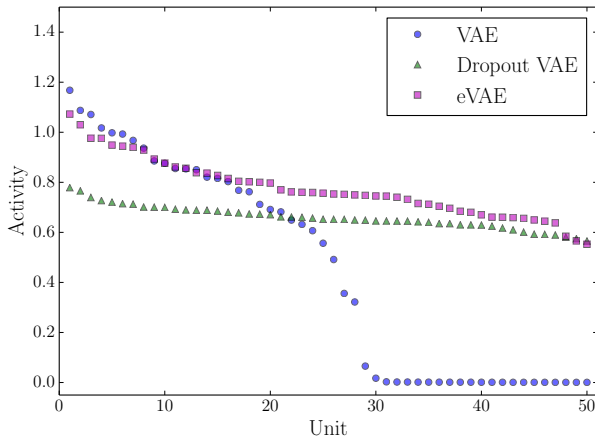

Figure 4: Adding dropout to a VAE (here, dropout rate 0.5 is shown) can prevent the model from pruning units, shown for MNIST. However, in contrast to eVAE, it uses the additional units to encode redundancy, not additional information, and therefore does not address the problem. Generation results are shown in Fig. 5.

## 3.2 TRAINING

The generative model and the recognition network are trained simultaneously, by minimizing $\mathcal{C}_{evae}$ in eq. 10.

For the stochastic continuous variable $\mathbf{z}$, we use the reparameterization trick as in VAE. The trick involves reparametrizing the recognition distribution in terms of auxiliary variables with fixed distributions. This allows efficient sampling from the posterior distribution as they are deterministic functions of the inputs and auxiliary variables.

For the discrete variable $y$, we cannot use the reparameterization trick. We therefore approximate $q(y|\mathbf{x})$ by a point estimate $y*$ so that $q(y|\mathbf{x}) = \delta(y = y*)$, where $\delta$ evaluates to 1 only if $y = y*$ and the best $y* = \arg\min \mathcal{C}_{evae}$. We also explored modeling $q(y|\mathbf{x}) = Cat(h(\mathbf{x}))$ as a discrete distribution with $h$ being a neural network. In this case, the backward pass requires either using REINFORCE or passing through gradients for the categorical sampler. In our experiments, we found that these approaches did not work well, especially when the number of possible values of $y$ becomes large. We leave this as future work to explore.

The recognition network first computes $\mu$ and $\phi$. It is then combined with the optimal $y*$ for each example, to arrive at the final posterior. The model is trained using a simple algorithm outlined in Algo. 1. Backpropagation with minibatch updates is used, with each minibatch constructed to be balanced with respect to epitome assignment.

---

**Algorithm 1** Learning Epitomic VAE

---

1: $\theta, \phi \leftarrow$ Initialize parameters
2: **for** until convergence of parameters $(\theta, \phi)$ **do**
3:     Assign each $\mathbf{x}$ to its best $y* = \arg\min \mathcal{C}_{evae}$
4:     Randomize and then partition data into minibatches with each minibatch having proportionate number of examples $\forall y$
5:     **for** k $\in$ numbatches **do**
6:         Update model parameters using $k^{th}$ minibatch consisting of $\mathbf{x}, y$ pairs
7:     **end for**
8: **end for**

---

## 4 EXPERIMENTS

We present experimental results on two datasets, MNIST (LeCun et al., 1998) and Toronto Faces Database (TFD) (Susskind et al., 2010). We show generation results that illustrate eVAE's ability to better utilize model capacity for modeling data variability, and then evaluate the effect of epitome choice and model complexity. Finally we present quantitative comparison with other models and qualitative samples from eVAE. We emphasize that in all experiments, we keep the weight of the KL term $\lambda = 1$ to evaluate performance under optimizing the true derived lower bound, without introducing an additional hyperparameter to tune.

We use standard splits for both MNIST and TFD. In our experiments, the encoder and decoder are fully-connected networks, and we show results for different depths and number of units of per layer. ReLU non-linearities are used, and models are trained using the Adam update rule (Kingma & Ba, 2014) for 200 epochs (MNIST) and 250 epochs (TFD), with base learning rate 0.001.

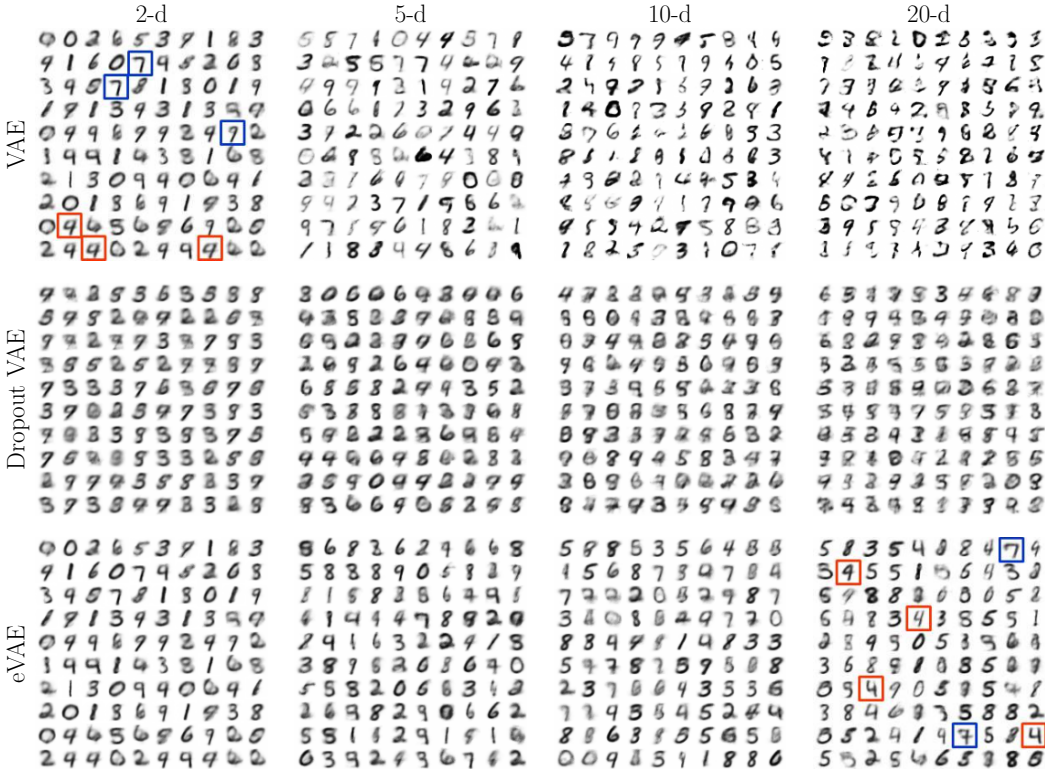

Figure 5: Generations from VAE, Dropout VAE, and eVAE models for different dimensions of latent variable **z**. Across each row are 2-d, 5-d, 10-d, and 20-d models. VAE generation quality (1st row) degrades as latent dimension increases, and it is unable to effectively use added capacity to model greater variability. Adding dropout to the VAE (2nd row) fails to solve the problem since additional units are used to encode redundancy, not additional information. eVAE (3rd row) overcomes the problem by modeling multiple shared subspaces, here 2-d (overlapping) epitomes are maintained as the latent dimension is increased. Learned epitome manifolds from the 20-d model are shown in Fig. 3. Boxed digits highlight the difference in variability that the VAE vs. eVAE model is able to achieve.

## 4.1 OVERCOMING OVER-PRUNING.

We first qualitatively illustrate the ability of eVAE to overcome over-pruning and utilize latent capacity to model greater variability in data. Fig. 5 compares generation results for VAE, Dropout VAE, and eVAE for different dimensions $D$ of latent variable **z**. With $D = 2$, VAE generates realistic digits but suffers from lack of diversity. When $D$ is increased to 5, the generation exhibits some greater variability but also begins to degrade in quality. As $D$ is further increased to 10 and 20, the degradation continues. As explained in Sec. 2.1, this is due to VAE's propensity to use only a portion of its latent units for modeling the training data and the rest to minimize the KL term. The under-utilization of model capacity means that VAE learns to model well only regions of the posterior manifold near training samples, instead of generalizing to model the space of possible generations. The effect of this is good reconstruction (examples are shown in Fig. 9) but poor generation samples.

Adding dropout to the latent variable **z** of the VAE (row 2 of Fig. 5) encourages increased usage of model capacity, as shown in Fig. 4 and the discussion in Sec. 2. However, due to the stochastic nature of dropout, the model is forced to use the additional capacity to encode redundancy in the representation. It therefore does not achieve the desired effect of encoding additional data variability, and furthermore leads to blurred samples due to the redundant encoding. Epitomic VAE addresses the crux of the problem by learning multiple specialized subspaces. Since the effective dimension of any example is still small, eVAE is able to model each subspace well, while encoding variability through multiple possibly shared subspaces. This enables the model to overcome over-pruning from which VAE suffered. Fig. 5 shows that as the dimension $D$ of **z** is increased

while maintaining epitomes of size $K = 2$, eVAE is able to model greater variability in the data. Highlighted digits in the 20-d eVAE show multiple styles such as crossed versus un-crossed 7, and pointed, round, thick, and thin 4s. Additional visualization of the variability in the learned 2-d manifolds are shown in Fig. 3. In contrast, the 2-d VAE generates similar-looking digits, and is unable to increase variability and maintain sample quality as the latent dimension is increased.

## 4.2 CHOICE OF EPITOME SIZE

We next investigate how the choice of epitome size, $K$, affects generation performance. We evaluate the generative models quantitatively through their samples by measuring the log-density with a Parzen window estimator Rifai et al. (2012). Fig. 6 shows the Parzen log-density for different choices of epitome size on MNIST, with encoder and decoder consisting of a single deterministic layer of 500 units. Epitomes are non-overlapping, and the results are grouped by total dimension $D$ of the latent variable $\mathbf{z}$. For comparison, we also show the log-density for VAE models with the same dimension $D$, and for mixture VAE (mVAE), an ablative version of eVAE where parameters are not shared. mVAE can also be seen as a mixture of independent VAEs trained in the same manner as eVAE. The number of deterministic units in each mVAE component is computed so that the total number of parameters is comparable to eVAE.

As we increase $D$, the performance of VAE drops significantly, due to over-pruning. In fact, the number of active units for VAE are 8, 22 and 24 respectively, for $D$ values of 8, 24 and 48. In contrast, eVAE performance increases as we increase $D$, with an epitome size $K$ that is significantly smaller than $D$. Table 1 provides more comparisons. This confirms the advantage of using eVAE to avoid overpruning and effectively capture data distribution.

eVAE also performs comparably or better than mVAE at all epitome sizes. Intuitively, the advantage of parameter sharing in eVAE is that each epitome can also benefit from general features learned across the training set.

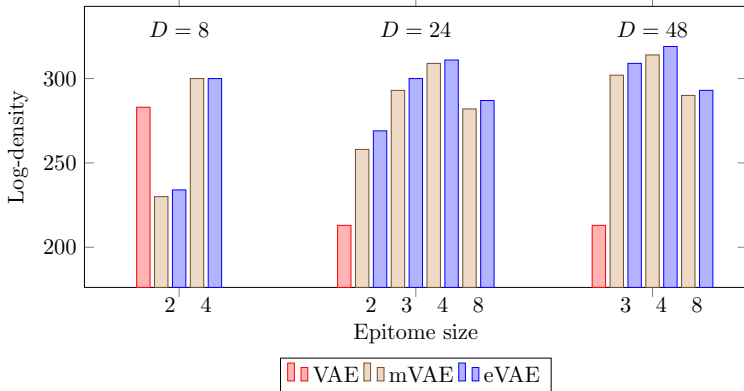

Figure 6: Epitome size vs. Parzen log-density (nats) on MNIST, grouped by different dimensions $D$ of latent variable $\mathbf{z}$. VAE performance for equivalent $D$ is shown for comparison, as well as mVAE (ablative version of eVAE without parameter sharing). For each $D$, the optimal epitome size is significantly smaller than $D$.

## 4.3 INCREASING COMPLEXITY OF ENCODER AND DECODER

Here, we would like to understand the role of encoder and decoder architectures on over pruning, and the generative performance. We control model complexity through number of layers $L$ of deterministic hidden units, and number of hidden units $H$ in each deterministic layer.

Table 1 shows the Parzen log-densities of VAE, mVAE and eVAE models trained on MNIST and TFD with different latent dimension $D$. For mVAE and eVAE models on MNIST, the maximum over epitomes of size $K = 3$ and $K = 4$ is used, and on TFD epitomes of size $K = 5$ are used. All epitomes are non-overlapping.

We observe that for VAE, increasing the number of hidden units $H$ (e.g. from 500 to 1000) for a fixed network depth $L$ has a negligible effect on the number of active units and performance. On the other hand, as the depth of the encoder and decoder $L$ is increased, the number of active units in VAE decreases though performance is still able to improve. This illustrates that increase in the complexity of the interactions through use of multiple

layers counteract the perils of the over-pruning. However, this comes with the cost of substantial increase in the number of model parameters to be learned.

In contrast, for any given model configuration, eVAE is able to avoid the over-pruning effect in the number of active units and outperform VAE. While both VAE and eVAE approach what appears to be a ceiling in generative performance with large models for MNIST, the difference between VAE and eVAE is significant for all TFD models.

Table 1 also shows results for mVAE, the ablative version of eVAE where parameters are not shared. The number of deterministic units per layer in each mVAE component is computed so that the total number of parameters is comparable to eVAE. While mVAE and eVAE perform comparably on MNIST especially with larger models (reaching a limit in performance that VAE also nears), eVAE demonstrates an advantage on smaller models and when the data is more complex (TFD). These settings are in line with the intuition that parameter sharing is helpful in more challenging settings when each epitome can also benefit from general features learned across the training set.

| | | $H = 500$ | | | $H = 1000$ | | |
| | | $L = 1$ | $L = 2$ | $L = 3$ | $L = 1$ | $L = 2$ | $L = 3$ |
|---|---|---|---|---|---|---|---|
| | | | | MNIST | | | |
| $D = 8$ | VAE | 283(8) | 292(8) | 325(8) | 283(8) | 290(8) | 322(6) |
| | mVAE | **300**(8) | 328(8) | **337**(8) | 309(8) | **333**(8) | **335**(8) |
| | eVAE | **300**(8) | **330**(8) | **337**(8) | **312**(8) | 331(8) | 334(8) |
| $D = 24$ | VAE | 213(22) | 273(11) | 305(8) | 219(24) | 270(12) | 311(7) |
| | mVAE | 309(24) | 330(24) | **336**(24) | 313(24) | **333**(24) | **338**(24) |
| | eVAE | **311**(24) | **331**(24) | **336**(24) | **317**(24) | **332**(24) | **336**(24) |
| $D = 48$ | VAE | 213(24) | 267(13) | 308(8) | 224(24) | 273(12) | 309(8) |
| | mVAE | 314(48) | **334**(48) | 336(48) | 315(48) | 333(48) | **337**(48) |
| | eVAE | **319**(48) | **334**(48) | **337**(48) | **321**(48) | **334**(48) | 332(48) |
| | | | | TFD | | | |
| $D = 15$ | VAE | - | 2173(15) | 2180(15) | - | 2149(15) | 2116(15) |
| | mVAE | - | 2276(15) | 2314(15) | - | **2298**(15) | 2343(15) |
| | eVAE | - | **2298**(15) | **2353**(15) | - | 2278(15) | **2367**(15) |
| $D = 25$ | VAE | - | 2067(25) | 2085(25) | - | 2037(25) | 2101(25) |
| | mVAE | - | 2287(25) | 2306(25) | - | **2332**(25) | 2351(25) |
| | eVAE | - | **2309**(25) | **2371**(25) | - | 2297(25) | **2371**(25) |
| $D = 50$ | VAE | - | 1920(50) | 2062(29) | - | 1886(50) | 2066(30) |
| | mVAE | - | 2253(50) | 2327(50) | - | 2280(50) | 2358(50) |
| | eVAE | - | **2314**(50) | **2359**(50) | - | **2302**(50) | **2365**(50) |

Table 1: Parzen log-densities in nats of VAE, mVAE and eVAE for increasing model parameters, trained on MNIST and TFD with different dimensions $D$ of latent variable $\mathbf{z}$. For mVAE and eVAE models on MNIST, the maximum over epitomes of size $K = 3$ and $K = 4$ is used, and on TFD epitomes of size $K = 5$ are used. All epitomes are non-overlapping. Across each row shows performance as the number of encoder and decoder layers $L$ increases for a fixed number of hidden units $H$ in each layer, and as $H$ increases. Number of active units are indicated in parentheses.

## 4.4 Comparison with other models

In Table 2 we compare the generative performance of eVAE with other models, using Parzen log-density. $\text{VAE}^-$, $\text{mVAE}^-$, and $\text{eVAE}^-$ refer to models trained using the same architecture as Adversarial Autoencoders, for comparison. Encoders and decoders have $L = 2$ layers of $H = 1000$ deterministic units. $D = 8$ for MNIST, and $D = 15$ for TFD. VAE, mVAE, and eVAE refer to the best performing models over all architectures from Table 1. For MNIST, the VAE model is $(L, H, D) = (3, 500, 8)$, mVAE is $(3, 1000, 24)$, and eVAE is $(3, 500, 48)$. For TFD, the VAE model is $(3, 500, 15)$, mVAE is $(3, 1000, 50)$, and eVAE is $(3, 500, 25)$.

We observe that eVAE significantly improves over VAE and is competitive with several state-of-the-art models, notably Adversarial Autoencoders. Samples from eVAE on MNIST and TFD are shown in Fig. 7.

| Method | MNIST(10K) | TFD(10K) |
|---|---|---|
| DBN | $138 \pm 2$ | $1909 \pm 66$ |
| Deep CAE | $121 \pm 1$ | $2110 \pm 50$ |
| Deep GSN | $214 \pm 1$ | $1890 \pm 29$ |
| GAN | $225 \pm 2$ | $2057 \pm 26$ |
| GMMN + AE | $282 \pm 2$ | $2204 \pm 20$ |
| Adversarial AE | $340 \pm 2$ | $2252 \pm 16$ |
| VAE$^-$ | $290 \pm 2$ | $2149 \pm 23$ |
| mVAE$^-$ | $333 \pm 2$ | $2298 \pm 23$ |
| eVAE$^-$ | $331 \pm 2$ | $2278 \pm 26$ |
| VAE | $325 \pm 2$ | $2180 \pm 20$ |
| mVAE | $\mathbf{338 \pm 2}$ | $2358 \pm 20$ |
| eVAE | $337 \pm 2$ | $\mathbf{2371 \pm 20}$ |

Table 2: Parzen log-densities in nats on MNIST and TFD. VAE$^-$, mVAE$^-$, and eVAE$^-$ refer to models trained using the same architecture as Adversarial Autoencoders, for comparison. VAE, mVAE, and eVAE refer to the best performing models over all architectures from Table 1.

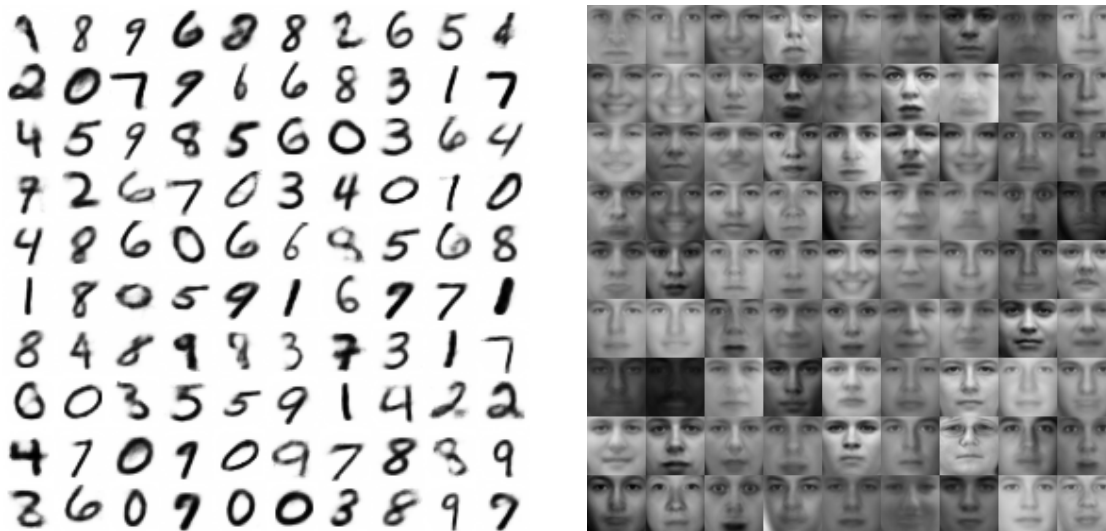

Figure 7: eVAE samples for MNIST (left) and TFD (right).

## 5 RELATED WORK

A number of applications use variational autoencoders as a building block. In Gregor et al. (2015), a generative model for images is proposed in which the generator of the VAE is an attention-based recurrent model that is conditioned on the canvas drawn so far. Eslami et al. (2016) proposes a VAE-based recurrent generative model that describes images as formed by sequentially choosing an object to draw and adding it to a canvas that is updated over time. In Kulkarni et al. (2015), VAEs are used for rendering 3D objects. Conditional variants of VAE are also used for attribute specific image generation (Yan et al., 2015) and future frame synthesis (Xue et al., 2016). All these applications suffer from the problem of model over-pruning and hence have adopted strategies that takes away the clean mathematical formulation of VAE. We have discussed these in § 2.1.

A complementary approach to the problem of model pruning in VAE was proposed in Burda et al. (2015); the idea is to improve the variational bound by using multiple weighted posterior samples. Epitomic VAE provides improved latent capacity even when only single sample is drawn from the posterior.

Methods to increase the flexibility of posterior inference are proposed in (Salimans et al., 2015; Rezende & Mohamed, 2016; Kingma et al., 2016). In Rezende & Mohamed (2016), posterior approximation is constructed by transforming a simple initial density into a complex one with a sequence of invertible transformations. In a similar vein, Kingma et al. (2016) augments the flexibility of the posterior through autoregression over projections of stochastic latent variables. However, the problem of over pruning still persists: for instance, Kingma et al. (2016) enforces a minimum information constraint to ensure that all units are used.

Related is the research in unsupervised sparse overcomplete representations, especially with group sparsity constraints *c.f.* (Gregor et al., 2011; Jenatton et al., 2011). In the epitomic VAE, we have similar motivations that enable learning better generative models of data.

# 6 CONCLUSION

This paper introduces Epitomic VAE, an extension of variational autoencoders, to address the problem of model over-pruning, which has limited the generation capability of VAEs in high-dimensional spaces. Based on the intuition that subconcepts can be modeled with fewer dimensions than the full latent space, epitomic VAE models the latent space as multiple shared subspaces that have learned specializations. We show how this model addresses the model over-pruning problem in a principled manner, and present qualitative and quantitative analysis of how eVAE enables increased utilization of the model capacity to model greater data variability. We believe that modeling the latent space as multiple structured subspaces is a promising direction of work, and allows for increased effective capacity that has potential to be combined with methods for increasing the flexibility of posterior inference.

# 7 ACKNOWLEDGMENTS

We thank the reviewers for constructive comments. Thanks to helpful discussions with Marc'Aurelio Ranzato, Joost van Amersfoort and Ross Girshick. We also borrowed the term 'epitome' from an earlier work of Jojic et al. (2003).

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

## 8 APPENDIX

### 8.1 EFFECT OF KL WEIGHT $\lambda$ ON RECONSTRUCTION

We visualize VAE reconstructions as the KL term weight $\lambda$ is tuned down to keep latent units active. The top half of each figure are the original digits, and the bottom half are the corresponding reconstructions. While reconstruction performance is good, generation is poor (Fig. 1). This illustrates that VAE learns to model well only regions of the posterior manifold near training samples, instead of generalizing to model well the full posterior manifold.

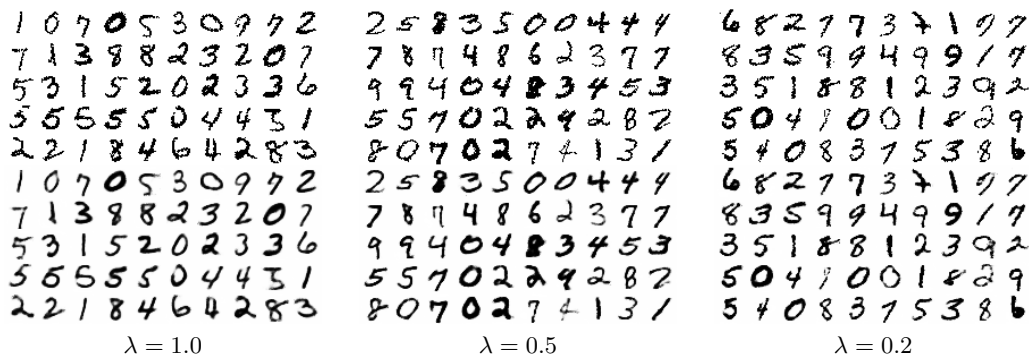

Figure 8: Reconstructions for a 50-d VAE with KL weight $\lambda = 1$, 0.5, and 0.2. The top half of each figure are the original digits, and the bottom half are the corresponding reconstructions.

## 8.2 EFFECT OF INCREASING LATENT DIMENSION ON RECONSTRUCTION

In § 4.1, Fig. 5 shows the effect of increasing latent dimension on generation for VAE, Dropout VAE, and eVAE models. Here we show the effect of the same factor on reconstruction quality for the models. The top half of each figure are the original digits, and the bottom half are the corresponding reconstructions. As the dimension of the latent variable **z** increases from 2-d to 20-d, reconstruction becomes very sharp (the best model), but generation degrades (Fig. 5). Dropout VAE has poorer reconstruction but still blurred generation, while eVAE is able to achieve both good reconstruction and generation.

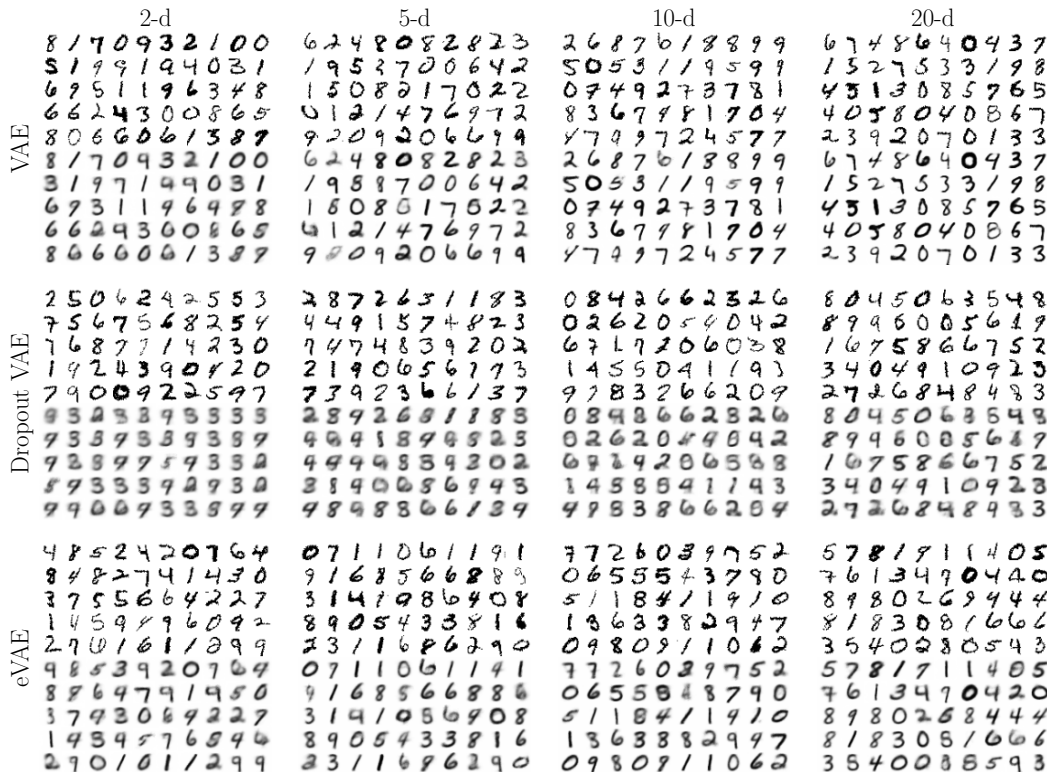

Figure 9: Reconstructions from VAE, Dropout VAE, and eVAE models for different dimensions of latent variable **z**. Across each row are 2-d, 5-d, 10-d, and 20-d models. The top half of each figure are the original digits, and the bottom half are the corresponding reconstructions. The eVAE models multiple shared subspaces by maintaining 2-d (overlapping) epitomes as the latent dimension is increased. eVAE is the only model that achieves both good reconstruction and generation.

### 8.3 EVALUATION METRIC FOR GENERATION

There have been multiple approaches for evaluation of variational autoencoders, in particular log-likelihood lower bound and log-density (using the Parzen window estimator, Rifai et al. (2012)). Here we show that for the generation task, log-density is a more appropriate measure than log-likelihood lower bound. Models are trained on binarized MNIST, to be consistent with literature reporting likelihood bounds. The encoder and decoder for all models consist of a single deterministic layer with 500 units.

Table 3 shows the log-likelihood bound and log-density for VAE and eVAE models as the dimension $D$ of latent variable $\mathbf{z}$ is increased. For VAE, as $D$ increases, the likelihood bound improves, but the log-density decreases. Referring to the corresponding generation samples in Fig. 11, we see that sample quality in fact decreases, counter to the likelihood bound but consistent with log-density. The reported VAE bounds and sample quality also matches Figs. 2 and 5 in Kingma & Welling (2014). On the other hand, eVAE log-density first decreases and then improves with larger $D$. We see that this is also consistent with Fig. 11, where eVAE samples for $D = 8$ are the most interpretable overall, and $D = 48$ improves over $D = 24$ but still has some degenerate or washed out digits. (Note that these models are consistent with Kingma & Welling (2014) but are not the best-performing models reported in our experiments.) Since our work is motivated by the generation task, we therefore use log-density as the evaluation metric in our experiments.

Intuitively, the reason why VAE improves the likelihood bound but generation quality still decreases can be seen in the breakdown of the bound into the reconstruction and KL terms (Table 3 and Fig. 10). The improvement of the bound is due to large improvement in reconstruction, but the KL becomes significantly worse. This has a negative effect on generation, since the KL term is closely related to generation. On the other hand, eVAE reconstruction improves to a lesser extent, but the KL is also not as strongly affected, so generation ability remains stronger overall. As a result of this, simply tuning the KL weight $\lambda$ in the training objective is insufficient to improve VAE generation, as shown in Fig. 1 in the main paper.

|  |  | Rec. term | KLD term | **Likelihood bound** | **Log-density** |
|---|---|---|---|---|---|
| | $D = 8$ | -89.4 | -16.6 | -106.0 | 278 |
| VAE | $D = 24$ | -61.1 | -29.3 | -90.4 | 152 |
| | $D = 48$ | -59.1 | -30.3 | -89.4 | 151 |
| | $D = 8$ | -110.1 | -9.6 | -119.7 | 298 |
| eVAE | $D = 24$ | -84.2 | -15.7 | -99.9 | 274 |
| | $D = 48$ | -82.8 | -14.2 | -97.0 | 284 |

Table 3: Likelihood bound and log-density for VAE and eVAE as dimension $D$ of latent variable $\mathbf{z}$ is increased. The encoder and decoder for all models consist of a single deterministic layer with 500 units. eVAE models have epitomes of size $K = 4$ for $D = 8$, and $K = 8$ for $D = 24$ and $D = 48$. The breakdown of the likelihood bound into reconstruction term and KLD term is also shown.

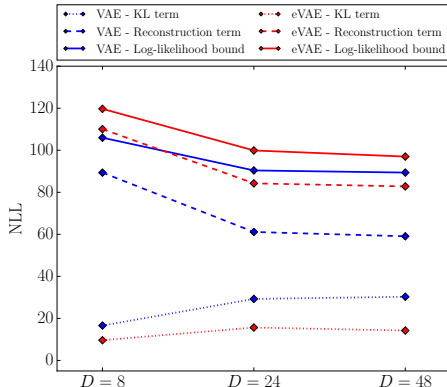

Figure 10: Likelihood bound for VAE and eVAE as $D$ increases (shown as NLL). VAE improvement of the bound is due to significant reduction of reconstruction error, but at high cost of KL, which is closely related to generation. eVAE improves reconstruction more moderately, but also maintains lower KL, and has stronger generation overall.

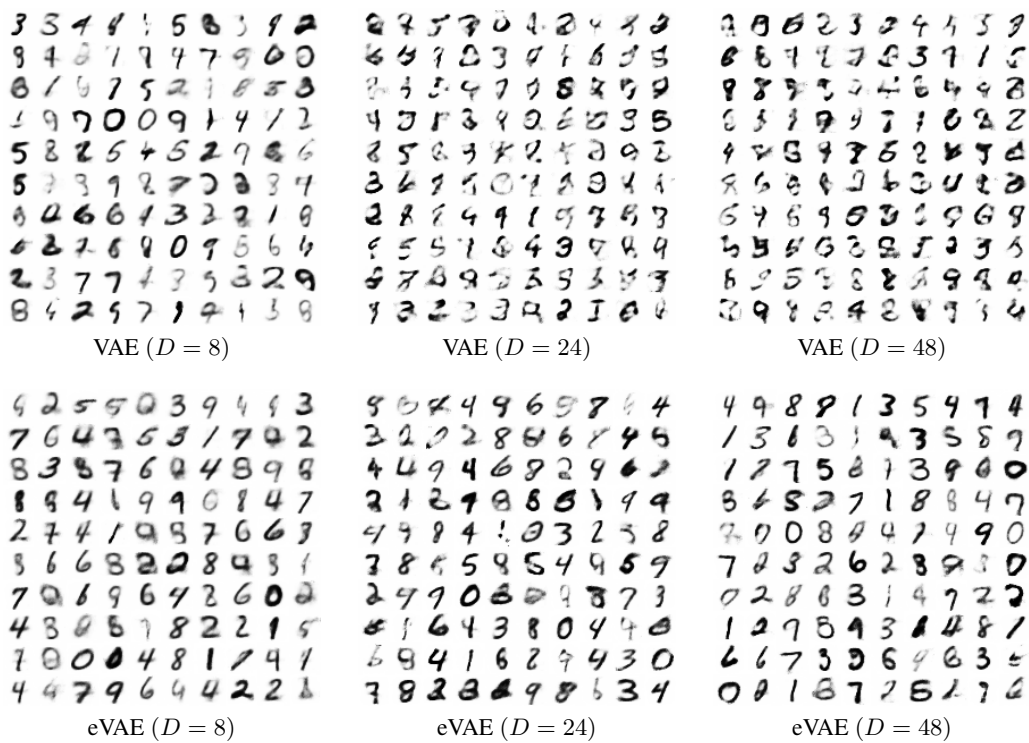

Figure 11: Generation samples for VAE and eVAE as dimension $D$ of latent variable $\mathbf{z}$ is increased. VAE sample quality decreases, which is consistent with log-density but not likelihood bound.

