# Peer review of "Epitomic Variational Autoencoders"

_ICLR 2017 — rejected_

[Official Review · AnonReviewer5 · rating 8 · confidence 5 · 14 Dec 2016]
**Addresses a fundamental limitation of the VAE. Great idea, well executed. Accept**

This paper proposes an elegant solution to a very important problem in VAEs, namely that the model over-regularizes itself by killing off latent dimensions. People have used annealing of the KL term and “free bits” to hack around this issue but a better solution is needed.
The offered solution is to introduce sparsity for the latent representation: for every input only a few latent distributions will be activated but across the dataset many latents can still be learned. 
What I didn’t understand is why the authors need the topology in this latent representation. Why not place a prior over arbitrary subsets of latents? That seems to increase the representational power a lot without compromising the solution to the problem you are trying to solve. Now the number of ways the latents can combine is no longer exponentially large, which seems a pity. 
The first paragraph on p.7 is a mystery to me: “An effect of this …samples”. How can under-utilization of model capacity lead to overfitting?
The experiments are modest but sufficient. 
This paper has an interesting idea that may resolve a fundamental issue of VAEs and thus deserves a place in this conference.

[Official Review · AnonReviewer4 · rating 4 · confidence 5 · 16 Dec 2016]
**skeptical of motivation and experiments**

This paper replaces the Gaussian prior often used in a VAE with a group sparse prior. They modify the approximate posterior function so that it also generates group sparse samples. The development of novel forms for the generative model and inference process in VAEs is an active and important area of research. I don't believe the specific choice of prior proposed in this paper is very well motivated however. I believe several of the conceptual claims are incorrect. The experimental results are unconvincing, and I suspect compare log likelihoods in bits against competing algorithms in nats.

Some more detailed comments:

In Table 1, the log likelihoods reported for competing techniques are all in nats. The reported log likelihood of cVAE using 10K samples is not only higher than the likelihood of true data samples, but is also higher than the log likelihood that can be achieved by fitting a 10K k-means mixture model to the data (eg as done in "A note on the evaluation of generative models"). It should nearly impossible to outperform a 10K k-means mixture on Parzen estimation, which makes me extremely skeptical of these eVAE results. However, if you assume that the eVAE log likelihood is actually in bits, and multiply it by log 2 to convert to nats, then it corresponds to a totally believable log likelihood. Note that some Parzen window implementations report log likelihood in bits. Is this experiment comparing log likelihood in bits to competing log likelihoods in nats? (also, label units -- eg bits or nats -- in table)

It would be really, really, good to report and compare the variational lower bound on the log likelihood!! Alternatively, if you are concerned your bound is loose, you can use AIS to get a more exact measure of the log likelihood. Even if the Parzen window results are correct, Parzen estimates of log likelihood are extremely poor. They possess any drawback of log likelihood evaluation (which they approximate), and then have many additional drawbacks as well.

The MNIST sample quality does not appear to be visually competitive. Also -- it appears that the images are of the probability of activation for each pixel, rather than actual samples from the model. Samples would be more accurate, but either way make sure to describe what is shown in the figure.

There are no experiments on non-toy datasets.

I am still concerned about most of the issues I raised in my questions below. Briefly, some comments on the authors' response:

1. "minibatches are constructed to not only have a random subset of training examples but also be balanced w.r.t. to epitome assignment (Alg. 1, ln. 4)."
Nice! This makes me feel better about why all the epitomes will be used.

2. I don't think your response addresses why C_vae would trade off between data reconstruction and being factorial. The approximate posterior is factorial by construction -- there's nothing in C_vae that can make it more or less factorial.

3. "For C_vae to have zero contribution from the KL term of a particular z_d (in other words, that unit is deactivated), it has to have all the examples in the training set be deactivated (KL term of zero) for that unit"
This isn't true. A standard VAE can set the variance to 1 and the mean to 0 (KL term of 0) for some examples in the training set, and have non-zero KL for other training examples.

4. The VAE loss is trained on a lower bound on the log likelihood, though it does have a term that looks like reconstruction error. Naively, I would imagine that if it overfits, this would correspond to data samples becoming more likely under the generative model.

5/6. See Parzen concerns above. It's strange to train a binary model, and then treat it's probability of activation as a sample in a continuous space.

6. "we can only evaluate the model from its samples"
I don't believe this is true. You are training on a lower bound on the log likelihood, which immediately provides another method of quantitative evaluation. Additionally, you could use techniques such as AIS to compute the exact log likelihood.

7. I don't believe Parzen window evaluation is a better measure of model quality, even in terms of sample generation, than log likelihood.

[Official Review · AnonReviewer3 · rating 5 · confidence 5 · 16 Dec 2016 (modified: 23 Jan 2017)]
**Interesting idea, experimental evidence doesn't confirm the presented story**

The paper presents a version of a variational autoencoder that uses a discrete latent variable that masks the activation of the latent code, making only a subset (an "epitome") of the latent variables active for a given sample. The justification for this choice is that by letting different latent variables be active for different samples, the model is forced to use more of the latent code than a usual VAE.
While the problem of latent variable over pruning is important and has been highlighted in the literature before in the context of variational inference, the proposed solution doesn't seem to solve it beyond, for instance, a mixture of VAEs. Indeed, a mixture of VAEs would have been a great baseline for the experiments in the paper, as it uses a categorical variable (the mixture component) along with multiple VAEs. The main difference between a mixture and an epitomic VAE is the sharing of parameters between the different "mixture components" in the epitomic VAE case.
The experimental section presents misleading results.
1. The log-likelihood of the proposed models is evaluated with Parzen window estimator. A significantly more accurate lower bound on likelihood that is available for the VAEs is not reported. In reviewer's experience continuous MNIST likelihood of upwards of 900 nats is easy to obtain with a modestly sized VAE.
2. The exposition changes between dealing with binary MNIST and continuous MNIST experiments. This is confusing, because these versions of the dataset present different challenges for modeling with likelihood-based models. Continuous MNIST is harder to model with high-capacity likelihood optimizing models, because the dataset lies in a proper subspace of the 784-dimensional space (some pixels are always or almost always equal to 0), and hence probability density can be arbitrarily large on this subspace. Models that try to maximize the likelihood often exploit this option of maximizing the likelihood by concentrating the probability around the subspace at the expense of actually modeling the data. The samples of a well-tuned VAE trained on binary MNIST (or a VAE trained on continuous MNIST to which noise has been appropriately added) tend to look much better than the ones presented in experimental results.
3. The claim that the VAE uses its capacity to "overfit" to the training data is not justified. No evidence is presented that the reconstruction likelihood on the training data is significantly higher than the reconstruction likelihood on the test data. It's misleading to use a technical term like "overfitting" to mean something else.
4. The use of dropout in dropout VAE is not specified: is dropout applied to the latent variables, or to the hidden layers of the encoder/decoder? The two options will exhibit very different behaviors.
5. MNIST eVAE samples and reconstructions look more like a more diverse version of 2d VAE samples/reconstructions - they are blurry, the model doesn't encode precise position of strokes. This is consistent with an interpretation of eVAE as a kind of mixture of smaller VAEs, rather than a higher-dimensional VAE. It is misleading to claim that it outperforms a high-dimensional VAE based on this evidence.

In reviewer's opinion the paper is not yet ready for publication. A stronger baseline VAE evaluated with evidence lower bound (or another reliable method) is essential for comparing the proposed eVAE to VAEs.

[Public Comment · Galin Georgiev · rating 6 · confidence 5 · 06 Jan 2017]
**Cool way to contain "over-sampling" of VAE. Wish there were non-toy experiements**

This paper is refreshing and elegant in its handling of "over-sampling" in VAE. Problem is that good reconstruction requires more nodes in the latent layers of the VAE. Not all of them can or should be sampled from at the "creative" regime of the VAE. Which ones to choose? The paper offers and sensible solution. Problem is that real-life data-sets like CIFAR have not being tried, so the reader is hard-pressed to choose between many other, just as natural, solutions. One can e.g. run in parallel a classifier and let it choose the best epitome, in the spirit of spatial transformers, ACE, reference [1]. The list can go on. We hope that the paper finds its way to the conference because it addresses an important problem in an elegant way, and papers like this are few and far between!

On a secondary note, regarding terminology: Pls avoid using "the KL term" as in section 2.1, there are so many "KL terms" related to VAE-s, it ultimately gets out of control. "Generative error" is a more descriptive term, because minimizing it is indispensable for the generative qualities of the net. The variational error for example is also a "KL term" (equation (3.4) in reference [1]), as is the upper bound commonly used in VAE-s (your formula (5) and its equivalent - the KL expression as in formula (3.8) in reference [1]). The latter expression is frequently used and is handy for, say, importance sampling, as in reference [2].

[1]

[Author Response · Serena Yeung · 16 Jan 2017]
**Updated paper**

We thank the reviewers for their helpful comments and suggestions.  Based on them, we have significantly updated the paper to include:

1. Comparisons with mVAE (mixture VAE), an ablated version of eVAE that does not share parameters between epitomes
2. Section 4.2 analyzing the effect of epitome size on generative performance
3. Section 4.3 analyzing the effect of encoder / decoder architectures on over-pruning and generative performance
4. Fig. 7 with qualitative samples from best eVAE models
5.  Section 8.3 in the Appendix comparing the effectiveness of likelihood lower bound and Parzen log-density as a metric for generation ability, and reporting numbers for both
6. Updating and clarifying Parzen experiments to be on MNIST and lower bound experiments to be on binarized MNIST, consistent with literature

We believe these updates address the reviewers' comments as well as provide additional insight.

[Final Decision · Program Chairs · 06 Feb 2017]
**ICLR committee final decision**

This paper addresses issues faced when using VAEs and the pruning of latent variables. Improvements to the paper have been accounted for and improved the paper, but after considering the rebuttal and discussion, the reviewers still felt that more was needed, especially in terms of applicability across multiple different data sets. For this reason, the paper is unfortunately not yet ready for inclusion in this year's proceedings.